# WhatsTrust: A Trust Management System for WhatsApp

**Fatimah Almuzaini** [1], **Sarah Alromaih** [2], **Alhanoof Althnian** [3] **and Heba Kurdi** [1,4,*]

1  Department of Computer Science, College of Computer and Information Sciences, King Saud University, Riyadh 11451, Saudi Arabia; 437204130@student.ksu.edu.sa
2  The National Center for Cybersecurity Technology, King Abdulaziz City for Science and Technology, Riyadh 11442, Saudi Arabia; salromaih@kacst.edu.sa
3  Department of Information Technology, College of Computer and Information Sciences, King Saud University, Riyadh 11451, Saudi Arabia; aalthnian@ksu.edu.sa
4  Department of Mechanical Engineering, Massachusetts Institute of Technology (MIT), Cambridge, MA 02139, USA
*  Correspondence: hkurdi@ksu.edu.sa

**Abstract:** Online communication platforms face security and privacy challenges, especially in broad ecosystems, such as online social networks, where users are unfamiliar with each other. Consequently, employing trust management systems is crucial to ensuring the trustworthiness of participants, and thus, the content they share in the network. WhatsApp is one of the most popular message-based online social networks with over one billion users worldwide. Therefore, it is considered an attractive platform for cybercriminals who spread malware to gain unauthorized access to users' accounts to steal their data or corrupt the system. None of the few trust management systems proposed in the online social network literature have considered WhatsApp as a use case. To this end, this paper introduces WhatsTrust, a trust management system for WhatsApp that evaluates the trustworthiness of users. A trust value accompanies each message to help the receiver make an informed decision regarding how to deal with the message. WhatsTrust is extensively evaluated through a strictly controlled empirical evaluation framework with two well-established trust management systems, namely EigenTrust and trust network analysis with subjective logic (TNA-SL) algorithms, as benchmarks. The experimental results demonstrate WhatsTrust's dominance with respect to the success rate and execution time.

**Keywords:** trust; social networks; subjective logic; WhatsApp

## 1. Introduction

Online social networks (OSNs) have become incredibly popular and an integral part of our daily activities. These networks build active communities for individuals who share common behaviors, interests, backgrounds, and/or friendships [1]. An OSN can be classified into different categories, such as social relation connections, messaging social networks, academic social networks, and professional social networks. These networks differ primarily in terms of the content shared by users and the network architecture. One of the most popular messaging social networks is WhatsApp, which is actively used by more than two billion users [2] to send and receive messages, including text messages, photos, voice messages, videos, and documents, free of charge, as an alternative to the short message service (SMS) [3].

WhatsApp has been a target to many cyber-attacks to distribute destructive content or gain unauthorized access to users' accounts [4]. For instance, WhatsApp users have recently been attacked by cybercriminals, who use Uniform Resource Locators (URLs) to deliver malware to gain unauthorized access to user accounts. Many WhatsApp users have fallen prey to what appears a hacking trend, which occurs as follows: a malicious user sends a message that contains catchy news, but with a

compromised URL. When the recipient clicks on the URL, the malicious user gains access to the recipient's account. They then start texting friends and family members of the recipient, pretending to be in a financial emergency and demanding a quick cash transfer or credit card details. This represents a real threat, as many unsuspecting users do respond to such requests. Such incidents call for a trust management system for WhatsApp, with which a user would receive a trust value along with the message and thus they can recognize harmful messages and block or report the sender.

Trust management systems are crucial components in online communication platforms to ensure the trustworthiness of participants, and thus, the content they share in the network. Trust and reputation are strongly related concepts although quite different [5]. On one hand, trust takes place between two parties, and a trust value reflects the opinion of one party on the trustworthiness of another party [6,7]. Basically, trust management systems classify network users, based on their behavior, into good users and malicious users. Good users deliver valuable content and give honest feedback, while malicious users inject the network with malignant content and give deceptive feedback. Malicious users may work individually or in groups (henceforth referred to as collectively malicious users) to achieve their goals. Collectively malicious users usually have more devastating harm to the network and are harder to detect [8–10].

Several trust management systems have been proposed in the literature for various platforms, such as peer-to-peer (P2P) networks [6,8,11–14], vehicular networks [15], wireless sensor networks [16–18], cloud [19,20], and OSNs [21–25]. However, designing an efficient trust management system for OSNs poses more challenges as OSNs are diverse, and thus, different factors need to be considered when designing a trust management system for a specific OSN. Recently, several studies have focused on developing trust management systems for generic OSNs [21–23], but only a few contributions have addressed trust in a specific OSN, such as Twitter [26,27].

This paper proposes WhatsTrust, a trust management system tailored to the WhatsApp social network based on subjective logic (SL). Subjective logic is a probabilistic logic that computes subjective opinions or beliefs about one or more domain elements [28]. Subjective logic has been exploited for this work due to its successful application in a wide range of domains, such as monitoring nodes behavior [15], measuring the quality of web services [7], and assessing nodes trustworthiness in P2P systems [14] and cloud computing [23].

WhatsTrust applies subjective logic [28] to compute the local trust value (opinion) between two WhatsApp users and proposes a novel model to compute the global trust value (reputation) for each user. Therefore, when a WhatsApp user receives a message, WhatsTrust provides them with a trust value, of the sender to help decide on how to react to the message accordingly. To the best of our knowledge, this is the first trust management system tailored for WhatsApp social network. However, the proposed approach can be adapted to any other messaging social network by modifying the system model to reflect that of the considered social network. Additionally, the proposed algorithm is scalable, since it avoids building a large opinion matrix, which is a common problem in SL-based systems. Furthermore, WhatsTrust is immune against collectively malicious users, which are usually hard to detect and cause devastating harm.

In summary, the main contributions of this paper are:

- A trust management system for one of the most popular social networking applications with high security risks, WhatsApp, which has not been targeted by any other previous work.
- A new approach to calculate a user's reputation in a way that marginalizes the harm of collectively malicious users.
- A well-controlled experimental framework for evaluating the proposed approach.

The rest of this paper is organized as follows: Section 2 reviews the related trust management systems in P2P networks and OSNs. Section 3 illustrates the system model while Section 4 introduces the WhatsTrust algorithms. The evaluation methodology and results are explained in Section 5. Finally, a summary with concluding remarks are provided in Section 6.

## 2. Literature Review

The main purpose of trust management systems is to help maintain trustworthy communication among users [29]. Trust management systems are popular in P2P networks due to the vulnerable nature of these networks. A classical trust management system for P2P file-sharing networks is EigenTrust [30], which assigns each peer a unique global trust value and effectively reduces the number of inauthentic file downloads in the network. However, the algorithm suffers from two main drawbacks. First, EigenTrust depends on the concept of pre-trusted peers, and hence, the system might be compromised if any pre-trusted peer is deceived by a malicious peer. Second, the algorithm cannot express negative trust values. Due to the special characteristics of EigenTrust, it has been subject to frequent improvements and several variants. For instance, HonestPeer [10] is one of the proposals to tackle the pre-trusted peers' problem in EigenTrust. HonestPeer moderates the dependency on pre-trusted peers by involving peers with high reputation values, i.e., honest peers, in the calculation of the reputation values of other peers. The negative trust value problem is addressed by the trust network analysis with subjective logic (TNA-SL) algorithm [9] which allows negative ratings to be propagated through the network.

The TNA-SL algorithm [9] is a rival algorithm that exploits graph theory and SL to manage trust between peers. It represents the relationship between peers as a directed serial parallel graph (DSPG) with no cycles. The algorithm expresses the trust value between two peers as a subjective opinion that comprises four factors: belief, disbelief, uncertainty, and base rate. The main advantage of this algorithm is the accuracy of its trust information. On the other hand, it suffers from losing some trust information during the pruning process to ensure an acyclic graph [13]. Besides, its running time is exponential due to long matrix chain multiplications required to locate a trusted peer in the indirect trust relationship between peers, which negatively reflects the algorithm scalability. Owing to TNA-SL advantages, many variations and enhancement to it have emerged. For instance, a heuristic is proposed in [31] to find a sub-optimal path with minimal information loss, while in [32], an edge-splitting approach is utilized to refine the trust network graph. The work in [23] addresses the runtime overheads and the scalability problems by introducing a lightweight algorithm, InterTrust, which replaces large trust matrices with simpler lists.

Compared to P2P, OSNs are considered a recent paradigm, and hence, trust in OSNs has been less studied. Reviews show that SL has also been widely used in OSNs to compute trust values of users. For example, the three-valued SL (3VSL), proposed in [21], calculates multi-hop trust values in arbitrary graphs using SL by distinguishing between posteriori and priori uncertainties. In Reference [19], an opinion walk algorithm is introduced which calculates trust based on 3VSL. Opinion walk starts the trustor's walk through the network using the breadth-first search technique and iteratively calculates users' trust values. A trust framework, SWTrust, which focuses on building trusted graphs from large OSNs is proposed in [33]. The proposed trusted graphs can be applied to existing trust algorithms to make them more efficient and practical.

Unlike many existing works, [34] proposes a trust model in OSNs based on observing disclosure of personal information rather than friendship. Trust evaluation model (T-OSN) is presented in [35], which considers the number of friends (degree) and contact frequency as two main factors to calculate trust values. In Reference [36], the authors proposed a trust and reputation framework for social networks, which takes into account the users relationships, the historic evolution of their reputations, and their profile similarity.

In Reference [26], a reputation-based credibility analysis model is proposed for Twitter users, where user reputation to score them based on popularity and sentimentality. In Reference [37], the authors presented CoRank, a method to evaluate the trustworthiness of users and tweets by analyzing user or tweet behaviors on Twitter. A framework for calculating trust on multiple heterogeneous social networks is introduced in [38] based on semantic web technology. The proposed approach applied weighted ordered weighted averaging data fusion technique to aggregate individual networks without distorting trust.

Artificial intelligence techniques, including machine learning and optimization, have been used to compute and evaluate trust in OSNs. For instance, Reference [39] proposed a multi-feature framework based on machine learning. It considers four trust features, which include profile-based trust, behavior-based trust, feedback-based trust, and link-based trust. In Reference [40] the author presented a metaheuristic algorithm based on the artificial bee colony (ABC) optimization for calculating the maximal trust and the trust route between any two users in an OSN. A dynamic algorithm for stochastic trust propagation in OSNs is proposed in [25] to infer the trust value between two indirectly connected users. The presented method utilized distributed learning automata to capture dynamic trust during the process of trust propagation and dynamically updated the trust paths. In Reference [24], a machine learning-based approach is followed to calculate the trust value for nodes in social networks. The approach first selects the best features then trains a logistic regression model to compute the node trust values.

Overall, previous studies that applied SL to address trust in OSNs are unscalable [22] due to the large computation overheads. While contributions that used machine learning models require large datasets, which are difficult to have and introduces space and runtime overheads. Above all, although some previous studies have considered specific OSN such as Twitter [26,37], no previous study has investigated trust in WhatsApp. On the other hand, WhatsApp has much wider reach than twitter [2], which attracts cybercriminals to spread malware threats via infected files or links. Based on the above, there is fundamental need for a trust management system for WhatsApp. This work seeks to fill that gap. This paper proposes WhatsTrust, an SL-based trust management system for WhatsApp.

## 3. System Model

### 3.1. Network Model

WhatsApp can be viewed as a P2P message-exchange network. Any user in the network can exchange messages with any other user or a group of users, irrespective of whether they are part of the user's contact list or not. As shown in Figure 1, WhatsTrust constructs a trust overlay network (ToN) on top of the WhatsApp OSN to provide a substrate for the structure of a large-scale system [41]. The ToN represents the underlying network as a directed graph. Each node in the graph denotes a WhatsApp user and each directed edge denotes the opinion of the source node in the destination node. For instance, a directed edge from node $n_i$ to node $n_j$ denotes the opinion of node $n_i$ in node $n_j$ based on the validity of previous messages received at node $n_i$ from node $n_j$.

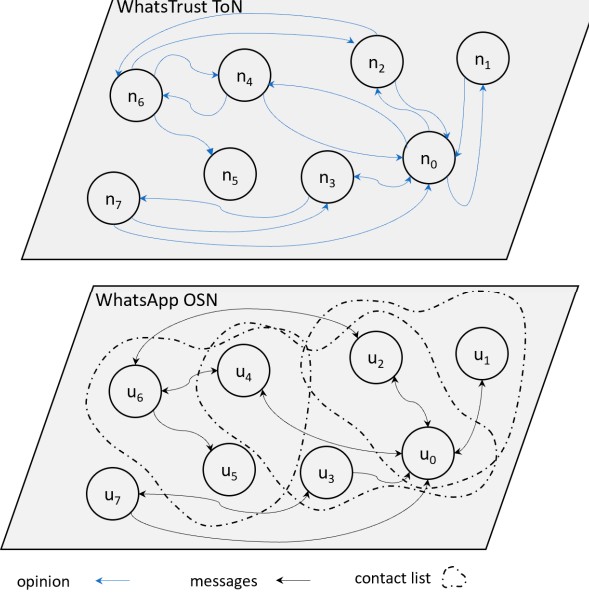

**Figure 1.** WhatsTrust trust overlay network (ToN).

### 3.2. User Models

Similar to P2P networks, WhatsTrust's nodes can be classified as follows:

- Good nodes: these are nodes that provide valid content and fair ratings about other nodes.
- Malicious nodes: these are nodes that intend to harm other nodes and distort their reputation. WhatsTrust considers two types of malicious nodes.
- Naive malicious nodes: these are malicious nodes that work individually to deliver invalid contents and unfair ratings to other nodes.
- Collectively malicious nodes: these are malicious nodes that form groups to distribute invalid contents and harm other nodes. A collectively malicious node gives positive ratings to the nodes within its group, to raise their reputations, and unfair negative ratings to all other nodes to distort their reputations.

### 3.3. Relationship Models

In WhatsTrust, different relationship models are considered based on user interactions in WhatsApp. Figure 2 illustrates these different relations for node $n_0$ (the top figure) based on the interaction of the corresponding WhatsApp user $u_0$ (the bottom figure). As shown in Section 4.1, each node maintains a local trust list, which keeps the trust information for all nodes with which he has previously interacted. More details on when a sender is added to a node's local trust list is provided in Section 4.2.1. The relationships between nodes are determined based on whether the sender node exists in the receiver node's local trust list and/or contact list or in his friends' contact lists. Below, we explain the relations based on the figure.

- Friend: a node $n_0$ considers another node ni (e.g., $n_2$, $n_3$, and $n_4$) a friend if *ni*'s mobile phone number (ID number) exists in $n_0$'s contact list.
- Acquaintance: A node $n_0$ considers another node (e.g., $n_7$) an acquaintance if $n_7$ exists in $n_0$'s local trust list but is not in the contact list. Acquaintances may also include group members.
- Friend of a friend (FoF): A node $n_0$ considers another node (e.g., $n_6$) a FoF if $n_6$ does not exist in $n_0$'s local trust list or contact list, but exists in his friends' contact list. As explained in Section 4.2.1, when a node $n_0$ receives a message from $n_6$ for the first time, $n_0$ checks with his friends (e.g., $n_4$) if n$_6$ is in their contact list. If so, $n_0$ considers $n_6$ to be a FoF.
- Blocked: a node $n_0$ may block another node (e.g., $n_1$), which prevents node $n_1$ from sending messages to $n_0$.
- Stranger: In all other cases, a node n$_0$ considers other nodes (e.g., $n_5$) to be strangers. This indicates that $n_5$ is not in $n_0$'s contact list or local trust list, nor is he in his friends' contact lists.

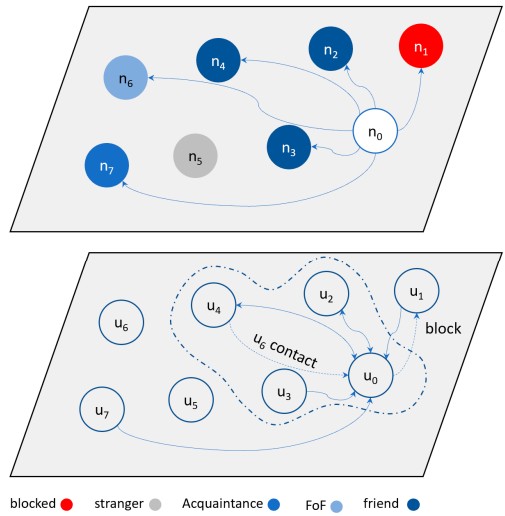

**Figure 2.** WhatsTrust relations model.

*3.4. Message-Exchange Models*

- Message-sending models:

  - A user may send a message to another user (one to one).
  - A user may send a message to a group of users using group chat (one to many). WhatsApp has a maximum group size of 256 users.
  - A user may broadcast a message to all contacts (one to all).
  - A user may send a message $m_1$ to request a reply message. When the recipient replies, $m_1$ is considered a request.

- Message-receiving models:

  - A user may not receive a message from a friend or any other user on the network unless the user is blocked.
  - Once a user opens a received message, the following actions may be taken:

    - Reply by responding to the message in a private chat with a user or a group chat with multiple users.
    - Forward the message to a friend, an acquaintance, or a group of users. Forwarded messages are labeled, which allows receiving users to identify whether the message is written by the sender.
    - Copy the content of a message and paste it in an outgoing message.
    - Delete a received message.
    - Star a received message by marking it with a star.

*3.5. Rating Model*

- Positive rating: WhatsTrust considers the following actions of a user who received a message (recipient) to be positive, and hence, assign the user who sends the message (sender) a positive rating.

  - The recipient user adds the sender to his contacts.
  - The recipient replies to the sender's message.
  - The recipient stars the message received from the sender.

- Negative rating: WhatsTrust considers the following actions of a recipient to be negative, and hence, assigns the sender a negative rating.

  - The recipient blocks the sender.
  - The recipient reports the sender.

## 4. WhatsTrust Algorithm Design

*4.1. System Architecture*

As shown in Figure 3, WhatsTrust comprises two main components: the system component and the node component. The following points explain each component in detail.

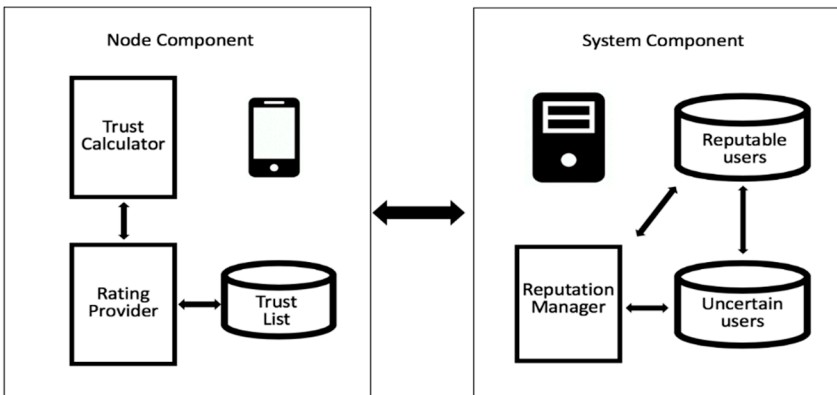

**Figure 3.** WhatsTrust architecture.

System component: this is a centralized component that manages the system registry and maintains two global lists: uncertain users and Reputable users lists. The former list includes rating information of newly joined users or users with questionable trustworthiness, while the latter list includes ratings of trusted users. Each list includes the user ID, positive rating, negative rating, and number of raters. The user ID is the user's mobile phone number. The positive rating is the total number of positive reactions sent by other users after interacting with the user with this ID, while the negative rating is that of negative reactions sent by others users after interacting with this user, based on the rating model explained in Section 3.5. Finally, the number of raters is the number of distinct users that contributed to the positive or negative ratings of this user. These values are important for the calculation of the global reputation of a user (as explained in Section 4.2), in order to prevent a single or a group of users from significantly impacting a user's reputation by repeatedly updating their ratings. These global lists provide valuable information to the new users who have recently joined the system, as they have no previous interactions with other users. The lists are also important to the old users to know the reputations of other users whom they, neither their friends, have previously interacted with. As the system needs to store the trust information of all WhatsApp users, maintaining two lists rather than one can contribute to a decrease in search time. The total length of the two lists is n, which is the total number of users in the system.

Node Component: This component comprises two main sub-components—trust calculator and rating provider and a local list, trust list, as shown in Figure 3. The trust calculator calculates the opinion about a sender node based on the rating received from the rating provider. The role of the rating provider is to decide from where to get the rating about a sender to the receiver node. This can be from the receiver's local trust list, friends or the global lists based on the relation between the two nodes. The local trust list of a node $n_i$ includes the trust information for all nodes with which $n_i$ has previously interacted. The purpose of maintaining a local trust list for each node is to provide a fast, efficient, and decentralized computation of trust opinions. Therefore, if a node has a history of interactions with another node, the node does not need to communicate with others to assess the trustworthiness of the other node. The list comprises three attributes: node ID, positive rating, and negative rating. The node ID is the mobile phone number of a node $n_j$ with which $n_i$ has previously interacted, the positive rating is the total number of previous positive ratings $n_i$ has given to $n_j$, and the negative rating is that of previous negative ratings.

### 4.2. How the Algorithm Works

This section presents the two algorithms for WhatsTrust that manages its main components, node algorithm and system algorithm. We explain each algorithm in more detail below.

### 4.2.1. Node Algorithm

Consider a user $n_i$, which has newly joined WhatsApp (for illustration purposes, $n_i$ takes the pronoun he). At system initialization, all contacts of $n_i$ are added to his local trust list with positive rating set to one, as they are considered as friends, and negative rating set to zero as no negative interactions have taken place yet. As $n_i$ interacts with other users, he updates their trust information in his trust list as well as the global lists.

When a node $n_i$ (recipient) receives a message from node $n_j$ (sender), $n_i$ computes his opinion about node $n_j$. As illustrated in Figure 4, WhatsTrust considers multiple cases based on whether $n_j$ is a friend or an acquaintance of $n_i$, a friend of a friend of $n_i$, a friend of multiple friends of $n_i$, or a stranger. If $n_j$ is a friend or an acquaintance of $n_i$, then $n_i$ would use his own local trust information in the trust list to compute his opinion about $n_j$ based on SL. Trust is modeled using four factors: belief (b), disbelief (d), uncertainty (u), and base rate ($\alpha$). Based on [9], the opinion of node $n_i$ about another node $n_j$, denoted by $\omega_{n_j}^{n_i}$, can be calculated based on Equation (1).

$$\omega_{n_j}^{n_i} = b + (\alpha \times u) \tag{1}$$

where b, d, u, and $\alpha \in [0, 1]$. Their values are computed based on the ratings of previous interactions between any two nodes according to Equations (2)–(4):

$$b = \frac{P}{(P + N + 2)} \tag{2}$$

$$d = \frac{N}{(P + N + 2)} \tag{3}$$

$$u = \frac{2}{(P + N + 2)} \tag{4}$$

where $P$ denotes the number of positive interactions and $N$ denotes that of negative interactions. The summation of b, d, and u is equal to one. The base rate $\alpha$ can take one of the two values based on whether the node is a friend or not, as shown in Equation (5):

$$\alpha = \begin{cases} 1.0 & \text{if user is a friend} \\ 0.5 & \text{otherwise} \end{cases} \tag{5}$$

If $n_j$ is not a friend or an acquaintance of $n_i$, then, he would send a request to his friends to check if they have interacted with $n_j$, and hence, have local trust information. Three scenarios are possible:

First, only one friend (e.g., $n_c$) responds, in which case $n_i$ would request the trust information from $n_c$ and compute the trust opinion about $n_j$ according to Equation (6) below.

$$\omega_{n_j}^{n_i:n_c} = \omega_{n_c}^{n_i} \otimes \omega_{n_j}^{n_c} \begin{cases} b_{n_j}^{n_i:n_c} = b_{n_c}^{n_i} \, b_{n_j}^{n_c} \\ d_{n_j}^{n_i:n_c} = d_{n_c}^{n_i} \, d_{n_j}^{n_c} \\ u_{n_j}^{n_i:n_c} = d_{n_c}^{n_i} + u_{n_c}^{n_i} + b_{n_c}^{n_i} \, u_{n_j}^{n_c} \\ \alpha_{n_j}^{n_i:n_c} = \alpha_{n_j}^{n_c} \end{cases} \tag{6}$$

Second, multiple friends (e.g., $n_c$ and $n_k$) respond. In this case, $n_i$ requests the trust information and computes the trust opinion about $n_j$ based on Equation (7) below.

$$\omega_{n_j}^{n_c n_k} = \omega_{n_j}^{n_c} \oplus \omega_{n_j}^{n_k} \begin{cases} b_{n_j}^{n_c \diamond n_k} = \dfrac{b_{n_j}^{n_c} u_{n_j}^{n_k} + b_{n_j}^{n_k} u_{n_j}^{n_c}}{\left(u_{n_j}^{n_c} + u_{n_j}^{n_k}\right) - \left(u_{n_j}^{c} u_{n_j}^{n_k}\right)} \\[12pt] d_{n_j}^{n_c \diamond n_k} = \dfrac{d_{n_j}^{n_c} u_{n_j}^{n_k} + d_{n_j}^{n_k} u_{n_j}^{n_c}}{\left(u_{n_j}^{n_c} + u_{n_j}^{n_k}\right) - \left(u_{n_j}^{n_c} u_{n_j}^{n_k}\right)} \\[12pt] u_{n_j}^{n_c \diamond n_k} = \dfrac{u_{n_j}^{n_c} u_{n_j}^{n_k}}{\left(u_{n_j}^{n_c} + u_{n_j}^{n_k}\right) - \left(u_{n_j}^{n_c} u_{n_j}^{n_k}\right)} \\[12pt] \alpha_{n_j}^{n_c \diamond n_k} = \alpha_{n_j}^{n_c} \end{cases} \tag{7}$$

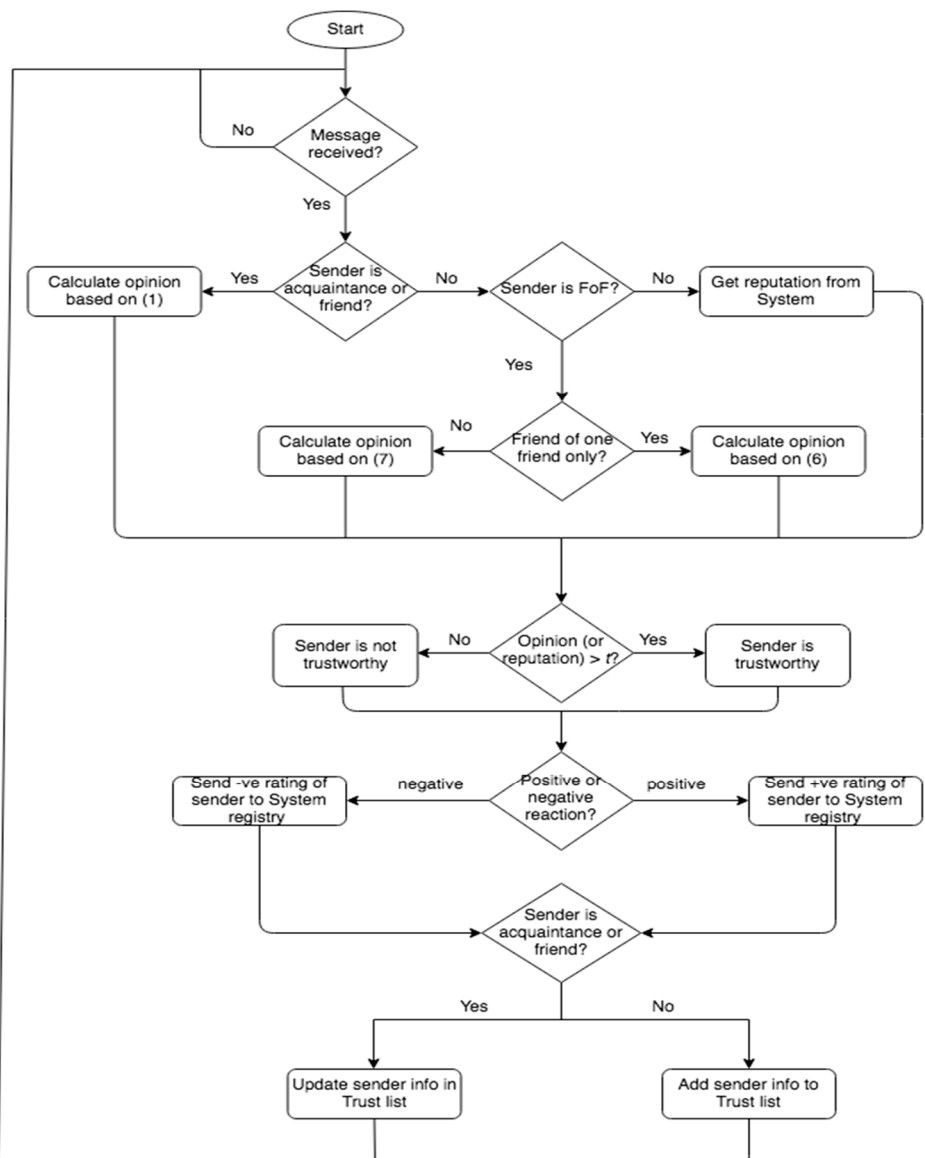

**Figure 4.** WhatsTrust node algorithm.

Third, if no friend responds within a predetermined period of time because they do not know $n_j$ or for any other communication issues, $n_j$ is considered a stranger. Therefore, $n_i$ requests $n_j$'s reputation value from the system, which, in turn, searches for the node's ratings in the reputable and uncertain user lists. Once it has been found, the system computes $n_j$'s reputation according to Equation (10), presented in Section 4.2.2.

Once the opinion or reputation of $n_j$ is available, that value is compared to a trust threshold $t$ to decide if $n_j$ is trustworthy as shown in Figure 4. The node $n_i$ is then in a position to informatively decide how to react to the message. If $n_i$ reacts positively to the message received by replying to the message, starring the message, and/or adding the sender $n_j$ to his contact list, a positive rating of $n_j$ is sent to the system, while if $n_i$ reacts negatively by blocking or reporting $n_j$, a negative rating is sent to the system, which accordingly updates the node reputation in one of the global lists. Additionally, $n_i$ updates the trust information of the sender locally. Therefore, if $n_j$ is a friend or acquaintance of $n_i$, the node will update his information in the local list, and if not, the node will be added to $n_i$'s local trust list. The node's information is updated to reflect whether the recipient's reaction is positive or negative. If the reaction is positive, the sender's positive rating is incremented by one in both the local and global lists, while if the reaction is negative, the sender's negative rating is incremented by one in both the local and global trust lists. The number of rating users in the global list is only incremented if this is the first time the sender has been rated by the recipient, which is an important constraint to eliminate the false ratings coming from collectively malicious nodes.

### 4.2.2. System Algorithm

The WhatsTrust system algorithm for updating nodes' global ratings and calculating their reputations is shown in Figure 5. The system algorithm is designed to take into account the behavior of collectively malicious nodes. As explained earlier in Section 1, collectively malicious nodes distribute harmful content and work together, to deceive other nodes, by giving each other positive ratings, to increase their reputation scores while negatively rates all other nodes. WhatsTrust considers two factors, namely the number of raters and the node reputation weight, to marginalize the harm of such behavior. Below, we explain the two factors in detail and how they are embedded within the algorithm.

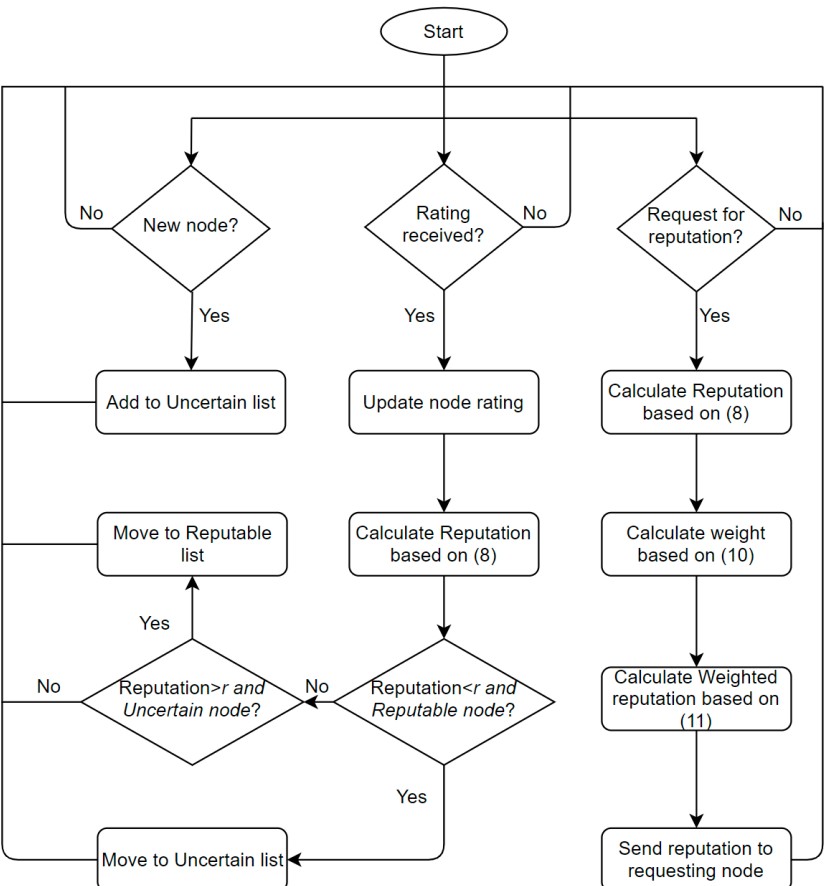

**Figure 5.** WhatsTrust system algorithm.

When a new user $n_i$ joins WhatsApp, he will be added to the uncertain users list. Once a rating of this node has been received from a node $n_j$, the system updates $n_i$ rating in the list by incrementing the node's positive or negative rating according to the rating received and increases the number of rating nodes if the node rates $n_i$ for the first time. The system computes $n_i$'s reputation based on Equation (8):

$$R_{ni} = \frac{P \times m}{(P + N)^2 + 2.0} \tag{8}$$

where $P$ is $n_i$'s positive rating, $N$ is his negative rating, and $m$ is the number of unique nodes who rated $n_i$. As shown in Equation (8), the positive rating of a node is multiplied by the number of rating nodes, and hence, a (good) node that receives his positive rating from a wide range of distinct nodes has higher reputation than a collectively malicious node that has the same positive rating but given by a smaller number of other collectively malicious nodes. After each update of the trust information of the node in the global lists, the system evaluates the node's candidacy to move from the current list to the other based on the following rule:

*If* $(R_{ni} \geq r) \wedge (n_i \in$ *Uncertain users list)*, *move $n_i$ to Reputable users list*
*else (if $R_{ni} < r$ ) $\wedge$ $(n_i \in$ Reputable users list)*, *move $n_i$ to Uncertain users list*

Therefore, if $n_i$ is reputable (i.e., $R_{ni} \geq r$) and currently exists in the uncertain users' list, it is moved to the reputable users list. On the other hand, if the node is uncertain (i.e., $R_{ni} < r$) and is currently stored in the reputable users list, the system moves the node to the uncertain users list. In our experiments, the value $r = 0.5$ gives the best results.

If the system receives a request for the reputation of a node $n_i$, it calculates the node's reputation according to Equation (8). Next, the system computes the node's reputation weight $w_{ni}$, which reflects the consensus of the ratings that a node receives. More weight is given to the rating of good nodes that have received consensus ratings from others, while less weight is given to the rating nodes that have received varying ratings from others. The reputation weight is calculated using Equation (9).

$$w_{ni} = \begin{cases} 0 & if \ P = 0, \ N = 0 \\ 1 & if \ P > 0, N = 0 \\ \frac{P-N}{P+N} & otherwise \end{cases} \tag{9}$$

where $P$ is $n_i$'s positive rating and $N$ is his negative rating. Then the system computes the $n_i$ weighted reputation, $wR_{ni}$, according to Equation (10), and then, sent to the node that requested it.

$$wR_{ni} = w_{ni} \times R_{ni} \tag{10}$$

## 5. Evaluation Methodology

### 5.1. Evaluation Framework

The performance of the WhatsTrust algorithm was evaluated by simulating some WhatsApp OSN scenarios using the quantitative trust management system (QTM) simulator for P2P networks [8]. WhatsTrust performance was compared with two well-established trust management algorithms, namely EigenTrust [30] and TNA-SL [9]. Additionally, the behaviors of the three algorithms were contrasted to the situation where no trust management system is utilized, the none algorithm, to serve as a baseline scenario.

To get a clear insight into the algorithm scalability, experiments were conducted with different number of populations; 100, 200 and 300 nodes and different number of transactions in the range between 2000 and 4000. Additionally, to examine the algorithm robustness under different number of malicious node, different percentages of malicious nodes were considered, including 20%, 40% and

60% of the total number of nodes taking into account both naïve and collective strategies of malicious nodes. We evaluated the WhatsTrust algorithm with respect to the following two performance metrics.

- Success rate: This metric is measured by dividing the total number of valid messages received from good nodes by that of transactions performed by good nodes, as shown in Equation (11).

$$\text{Success rate} = \frac{\text{number of authentic messages received by good nodes}}{\text{total number of messages received by good nodes}} \times 100 \qquad (11)$$

- Execution time: This metric measures the simulation running time of each algorithm: WhatsTrust, EigenTrust, TNA-SL and the none scenario. It is important to note that all algorithms were executed on the same computer cluster at the same load, so this measure can give a clear insight on the computing complexity and overheads of each algorithm.

### 5.2. Results and Discussion

This section presents the experimental results of the WhatsTrust algorithm in terms of the success rate and execution time. We compare the obtained results with EigenTrust [30], TNA-SL [9], and none algorithms. All reported results are averaged over ten simulation runs on the SANAM computer cluster [42].

5.2.1. Success Rate

The performance of the WhatsTrust algorithm with respect to the success rate, when nodes perform 2000, 3000 and 4000 transactions is shown in Figures 6–8. For each figure, the performance is presented when malicious nodes follow the naïve strategy (first row) and collective strategy (second row). Each sub-figure presents the success rate for WhatsTrust, EigenTrust, TNA-SL and none (as a baseline) over an increasing value of percentage of malicious nodes and total number of nodes.

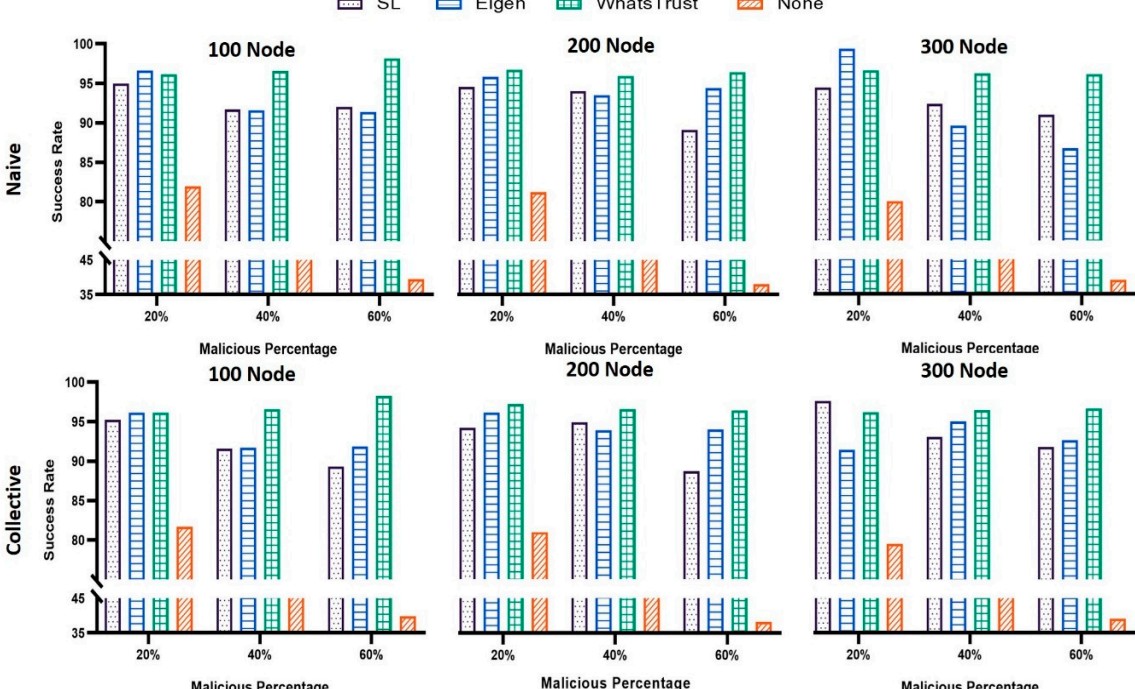

**Figure 6.** Success rate for 2000 transactions.

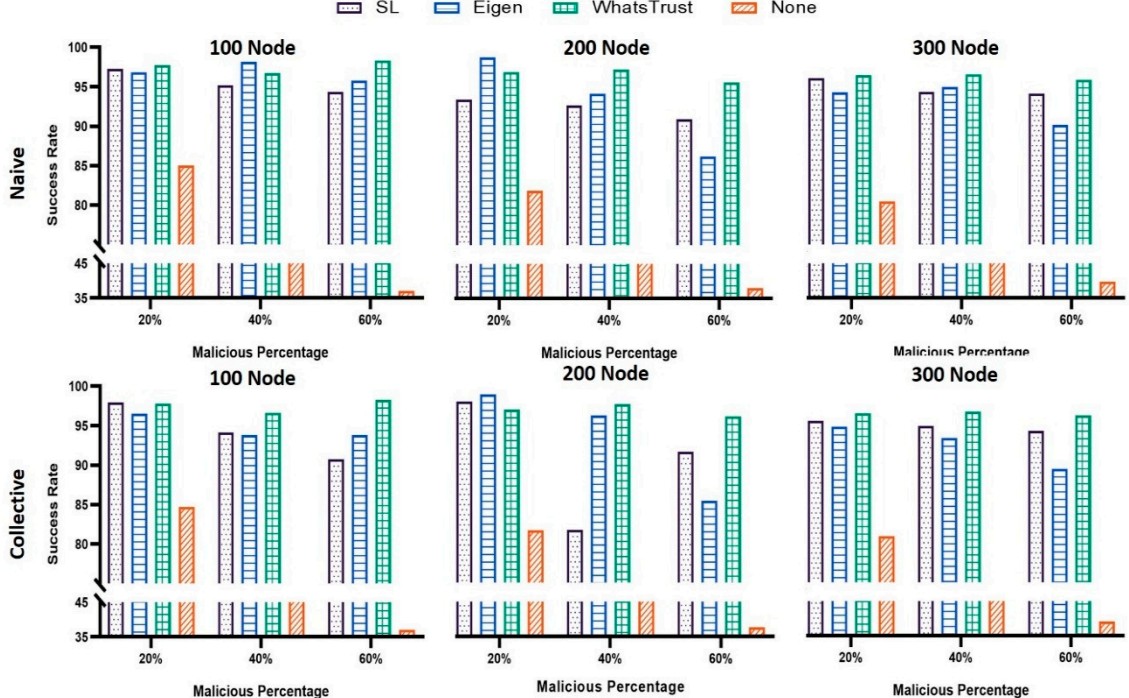

**Figure 7.** Success rate for 3000 transactions.

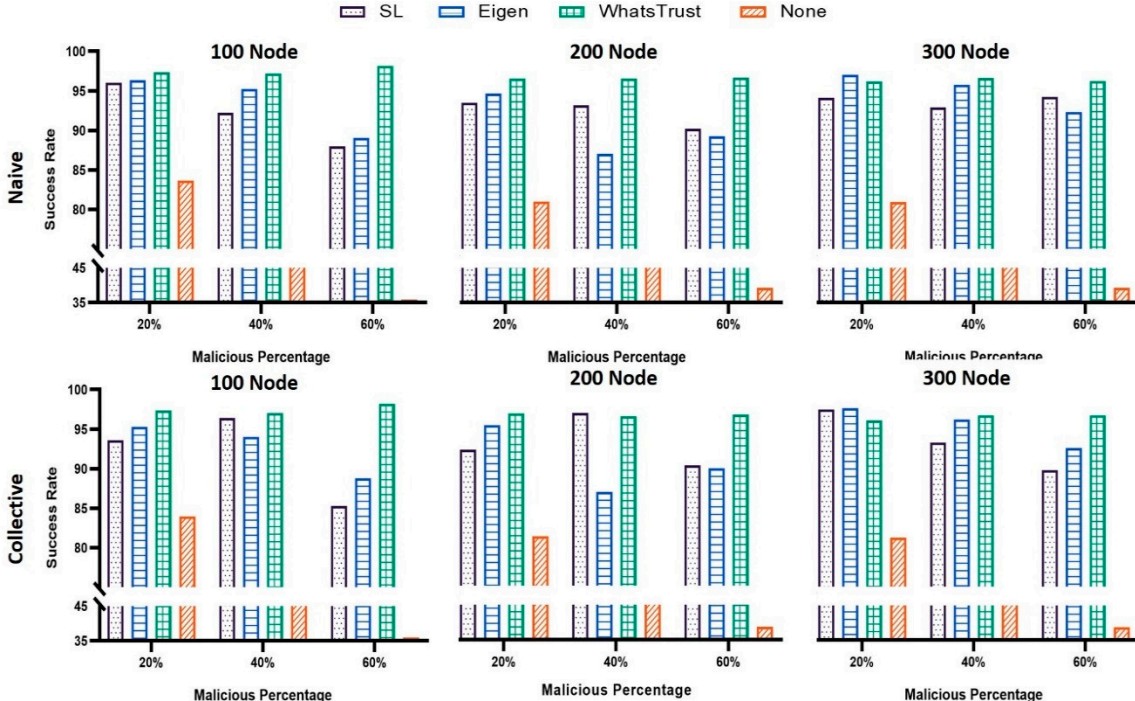

**Figure 8.** Success rate for 4000 transactions.

Success rate is measured for good nodes as the total number of valid messages received by good nodes divided by the total number of messages received. A high success rate suggests that most messages received by good nodes are valid because they can identify malicious nodes using the trust management algorithm, while a low success rate indicates that good nodes cannot identify malicious nodes, and hence, receive many invalid messages.

The figures suggest that WhatsTrust (for both naïve and collective strategies) outperforms the benchmarks, EigenTrust and TNA-SL in most of the scenarios presented in Figures 6–8 especially those of large number of nodes (more than 200 nodes) and high percentages of malicious nodes (above 20%) which clearly indicates the algorithms scalability and robustness. In fact, WhatsTrust exhibits reliability as it maintains a high success rate (>95%) across all scenarios, while the benchmarks show fluctuating performance where the success rate may drop significantly in some scenarios. For instance, as the percentage of malicious nodes increases, the success rate of EigenTrust decreases remarkably when 200 nodes performed 3000 transactions in Figure 7. In the same scenario, the success rate of TNA-SL drops significantly when 40% of nodes are malicious following the collective strategy, compared to the success rate when only 20% of the nodes are malicious.

Similar observations can be made by comparing scenarios of increasing number of transactions in Figures 6–8. For EigenTrust, consider the scenario of 200 nodes performing a minimum number of transactions (i.e., 2000 transactions in Figure 6) and a maximum number of transactions (i.e., 4000 transactions in Figure 8) with 40% malicious nodes. In the former case, both WhatsTrust and EigenTrust have a success rate of ≥5%. WhatsTrust maintains a similar rate in the latter scenario, while the performance of EigenTrust drops. Similar trends can be seen for TNA-SL and WhatsTrust in the scenario of 100 nodes performing 2000 (Figure 6) and 4000 transactions (Figure 8) with 60% malicious nodes.

### 5.2.2. Execution Time

We have measured the simulation run time, of each algorithm each scenario under both naïve and collective strategies. The results are presented in Figure 9. Each sub-figure corresponds to a specific number of nodes and shows the execution time, measured in seconds, required to perform an increasing number of transactions.

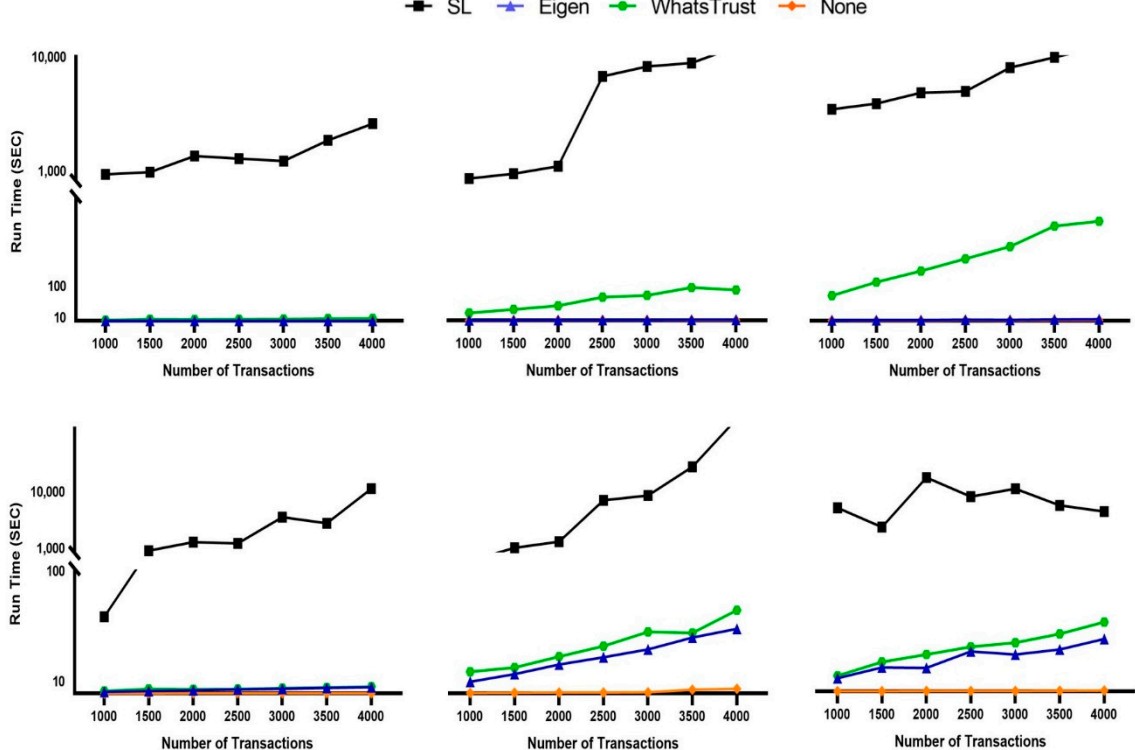

**Figure 9.** Execution times for EigenTrust, trust network analysis with subjective logic (TNA-SL), WhatsTrust, and none.

The figures demonstrate the speed efficiency and low complexity of the proposed algorithm, WhatsTrust, where it shows almost a linear running time (at n) in all scenarios. Both WhatsTrust and EigenTrust exhibit similar performance especially when the collectively malicious nodes are considered. It is only for naïve strategy when WhatsTrust presents a slightly longer running time than EigenTrust. On the other hand, the TNA-SL algorithm presents a significantly higher running time than all of three algorithms in all scenarios due to the expensive long chains of matrix multiplications needed by this algorithm when a message is received from a FoF.

## 6. Conclusions

This paper presents WhatsTrust, an SL-based trust management system. The proposed algorithm is tailored for one of the most popular message-based social networks, WhatsApp. WhatsTrust builds a ToN on top of the WhatsApp OSN, which represents the underlying network as a directed graph in which each node denotes a WhatsApp user and each directed edge from one node to another denotes his opinion. WhatsTrust comprises two algorithms: node and system. The node algorithm allows users who receive a message to compute their own opinions about the sender based on their past interactions in a decentralized manner, using SL. The system algorithm introduces an approach that allows the system to compute reputation for nodes based on global ratings, so that the users can assess the trust of unknown senders. WhatsTrust is designed to detect the behavior of collectively malicious nodes, thus assigning them low reputation.

The experimental results suggest that WhatsTrust outperforms two well-established trust management systems, namely EigenTrust and TNA-SL, in terms of success rate. Furthermore, WhatsTrust proves to be reliable because it maintains a high success rate (>95%) across all scenarios of increasing number of nodes and transactions as well as percentage of malicious nodes. In addition, the algorithm performs comparably with both naïve and collective strategies, which shows that it can reduce the chance of collectively malicious nodes deceiving other nodes. Moreover, WhatsTrust outperforms TNA-SL in terms of execution time and performs comparably to EigenTrust when focusing on collectively malicious nodes.

Although WhatsApp is a closed system with no Application Programming Interface (API) for external access, the authors recognizes the urgent need for a trust management system for WhatsApp. Hence, the proposed system is meant to be a proof of concept, which can be adopted by WhatsApp. The system component can be implemented as increment to the main application and the node component can be offered as an application update to the users. As a future work, it is intended to extend WhatsTrust to include not only user-based trust but also message-based trust management systems as due to the extremely large number of active WhatsApp users, many infected messages may be forwarded mistakenly by good users, and hence, recipients trust the content of the received message.

**Author Contributions:** Conceptualization, F.A. and A.A.; Data curation, S.A.; Formal analysis, F.A., A.A. and H.K.; Funding acquisition, H.K.; Methodology, H.K.; Software, F.A.; Visualization, S.A.; writing—original draft, F.A.; writing—review & editing, A.A. and H.K. All authors have read and agreed to the published version of the manuscript.

**Funding:** This research received no external funding.

**Acknowledgments:** This research was supported by a grant from Researchers Supporting Unit, project number (RSP-2020/204), King Saud University, Riyadh, Saudi Arabia.

**Conflicts of Interest:** The authors declare no conflict of interest.

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
