# Peer review of "WhatsTrust: A Trust Management System for WhatsApp"

_electronics, doi:10.3390/electronics9122190_

Round 1
Reviewer 1 Report
A subjective logic-based whatsTrust is proposed for WhatsApp user. Two algorithms are proposed for node and system. This reviewer is not convinced of the justification and the system model.
In open social networks such as Facebook and tweeter, trust models are mostly used to trust on the information posted by the users or determine the credibility of information (Fake news). Unlike tweeter and Facebook, Whatsapp is a restrictive online social network, where you can text/communicate someone only when either you know their number or already added them. A number not added in the list can be of acquaintances, friends of friends and strangers. How relationships will be determined?
Additionally, the threat scenario is of messaging a malware (or a link), which leads to clicking and downloading of the malware. At the system level, it makes sense to stop the advertisement of messages from unknown numbers with a bad reputation but leaving this to the end-user to make a decision would not be justified.
The authors should justify why a trust model is needed for Whatsapp? What is the similarity of WhatsApp to tweeter?
Some errors need attention.
Line 16-20 of abstracts are not relevant to the paper. 5G has nothing to do with Whatsapp and the same is for the introduction part of the paper.
Line 158 --> in OSNs are either are unscalable
173 --> node ni
172 -175 --> rephrase, difficult to read
227-228 --> formating errors in bullets
Author Response
We thank the reviewers for their reading of the manuscript and for their constructive comments. We have taken the comments into consideration to improve the quality of the manuscript. Please find below a point by point response to each comment.
Response to Reviewer 1 Comments
Major Comments
- Comment 1:
- A subjective logic-based whatsTrust is proposed for WhatsApp user. Two algorithms are proposed for node and system. This reviewer is not convinced of the justification and the system model.
- In open social networks such as Facebook and tweeter, trust models are mostly used to trust on the information posted by the users or determine the credibility of information (Fake news). Unlike tweeter and Facebook, Whatsapp is a restrictive online social network, where you can text/communicate someone only when either you know their number or already added them. A number not added in the list can be of acquaintances, friends of friends and strangers. How relationships will be determined?
- Additionally, the threat scenario is of messaging a malware (or a link), which leads to clicking and downloading of the malware. At the system level, it makes sense to stop the advertisement of messages from unknown numbers with a bad reputation but leaving this to the end-user to make a decision would not be justified.
The authors should justify why a trust model is needed for Whatsapp? What is the similarity of WhatsApp to tweeter?
Response 1: There are multiple related points raised in this comment. We have numbered them for easy reference. The following is a response to the first and third points since they are related. A response to the second point is provided afterwards.
The proposed system is a user-based trust management system for WhatsApp. The notion of user-based trust is derived from psychology and sociology, which indicates the subjective expectation that an entity has about another’s future behaviour. Unlink the system’s trust, the user’s trust systems are used to help users make informed decisions regarding how to interact with each other by evaluating the trustworthiness of participants, and thus, the content they share in the network. Therefore, a trust value reflects the opinion of one party on the trustworthiness of another party based on their previous direct or indirect interactions. This type of trust systems is very popular in OSNs and P2P file-sharing networks, and has been applied extensively as shown in the literature review section.
Both WhatsApp and Twitter are considered as online social networks and have been subject to many cyberattacks. However, WhatsApp is a messaging solution to facilitate communication between individuals, while Twitter is a microblog. This difference calls for a different trust management framework for each of them. With that being said, and considering the fact that WhatsApp has triple the users of Twitter, a trust management system is extremely needed for WhatsApp to help the users make informed decisions regarding how to deal with the received messages.
A system model has been designed to capture WhatsApp’s underlying network model, relationships models, message-exchange models, and rating model. Although the proposed approach has been applied to WhatsApp as a usecase, it can be adapted to any other messaging social network by modifying the system model to reflect that of the considered social network. In order to address this comment in the manuscript, the following sentence has been added to the introduction: “To the best of our knowledge, this is the first trust management system tailored for WhatsApp social network. However, the proposed approach can be adopted to any other messaging social network by modifying the system model to reflect that of the considered social network.”
The following is a response to the second point:
Relationships are determined according to the proposed relationship models (section 3.3) in line 210. As presented in the section, the relationships are determined as follow:
- Friend: A node n0 considers another node ni (e.g., n2, n3, and n4) a friend if n0 adds ni’s mobile phone number (ID number) to its contacts.
- Acquaintance: A node n0 considers another node (e.g., n7) an acquaintance if n0 receives a message form n7 but does not add its mobile phone number (ID number) to the contact list. Acquaintances may also include group members.
- Friend of a friend (FoF): A node n0 may receive the contact information of another node (e.g., n6) from a friend (e.g., n4). In this case, the trust information of n6 will also be shared. The user n0 considers n6 to be a friend of a friend.
- Blocked: A node n0 may block another node (e.g., n1), which prevents node n1 from sending messages to n0.
- Stranger: In all other cases, a node n0 considers other nodes (e.g., n5) to be strangers.
Minor Comments
- Comment 2: Line 16-20 of abstracts are not relevant to the paper. 5G has nothing to do with Whatsapp and the same is for the introduction part of the paper.
Response 2: The indicated lines of the abstract have been revised to reflect the paper. The lines 16-20 now reads: “Online communication platforms face security and privacy challenges, especially in broad ecosystems, such as online social networks, where users are unfamiliar with each other. Consequently, employing trust management systems is crucial to ensuring the trustworthiness of participants, and thus, the content they share in the network. WhatsApp is one of the most popular message-based online social networks with over one billion users worldwide.”
Similarly, the first few lines of the introduction related to 5G have been deleted as requested. We have also removed 5G from the keywords.
- Comment 3: Line 158 --> in OSNs are either are unscalable
Response 3: “are either” have been removed as suggested.
- Comment 4: 173 --> node ni
Response 4: ni has been deleted to improve readability of the sentence.
- Comment 5: 172 -175 --> rephrase, difficult to read
Response 5: The sentence has been rephrased and divided into three sentences. It now reads as follow: “The ToN represents the underlying network as a directed graph. Each node in the graph denotes a WhatsApp user and each directed edge denotes the opinion of the source node in the destination node. For instance, a directed edge from node ni to node nj denotes the opinion of node ni in node nj based on the validity of previous messages received at node ni from node nj.”
- Comment 6: 227-228 --> formating errors in bullets
Response 6: The formatting of the bullets has been fixed.

Reviewer 2 Report
As the title implies, the authors propose a trust management system for WhatsApp. The idea is rather interesting and addresses a real-world problem. My main issue with the proposed solution is where and by whom is this system expected to be implemented and used. While I understand the value, it is not straightforward who will be able to use it as to the best of my knowledge there is no mechanism to import these messages and process them from the user side. In essence, the proposed platform sits on a layer above Whatsapp which for instance is not available for, e.g. current available Whatsapp architecture and mobile app. Therefore, how is the needed information going to be collected? As a minor comment, while I understand the focus is on Whatsapp, I don't see the reason why it shouldn't be used on other similar software (e.g. Telegram, Viber etc.)
Author Response
We thank the reviewers for their reading of the manuscript and for their constructive comments. We have taken the comments into consideration to improve the quality of the manuscript. Please find below a point by point response to each comment.
Response to Reviewer 2 Comments
- Comment 7: As the title implies, the authors propose a trust management system for WhatsApp. The idea is rather interesting and addresses a real-world problem. My main issue with the proposed solution is where and by whom is this system expected to be implemented and used. While I understand the value, it is not straightforward who will be able to use it as to the best of my knowledge there is no mechanism to import these messages and process them from the user side. In essence, the proposed platform sits on a layer above Whatsapp which for instance is not available for, e.g. current available Whatsapp architecture and mobile app. Therefore, how is the needed information going to be collected?
Response 7:
“Although WhatsApp is a closed system with no API for external access, the authors recognizes the urgent need for a trust management system for WhatsApp. Hence, the proposed system is meant to be a proof of concept, which can be adopted by WhatsApp. The system component can be implemented as increment to the main application and the node component can be offered as an application update to the users.” The previous paragraph has been added to the conclusion to address this comment.
Comment 8: As a minor comment, while I understand the focus is on WhatsApp, I don't see the reason why it shouldn't be used on other similar software (e.g. Telegram, Viber etc.)
Response 8:
Although the proposed approach has been applied to WhatsApp as a usecase, it can be adapted to any other messaging online social network by modifying the system model to reflect that of the considered social network. In order to address this comment in the manuscript, the following sentence has been added to the introduction: “To the best of our knowledge, this is the first trust management system tailored for WhatsApp social network. However, the proposed approach can be adopted to any other messaging social network by modifying the system model to reflect that of the considered social network.”

Round 2
Reviewer 1 Report
Thank you for the response and modification to the original manuscript.
The response to the relationship is unsatisfactory. In WhatsApp, a person can text only for a number already in the friend's list. If a person number is not added in the list and a person texts for the first time then WhatsApp will flag it. The question remains the same that how it will be categorized that a number not added in the list is of a stranger, acquaintances or friends of friends. How the relationship decided? In Figure 4, there are some messaging going back and forth between friends to check the number reputation, what happens if friends are unable to communicate due to networking or any other issue?
A stranger categorisation is straightforward that it is not on the list. If a stranger text for the first time then is the number promoted from stranger to acquaintance upon reply or friend of a friend?
Do the authors categorise the acquaintance as to whom a message is previously sent and the number is promoted from stranger to acquaintance?
Authors need to consider different use cases of those relationships as the whole trust model is dependent on the relationships.
Author Response
- Comment 1:
Thank you for the response and modification to the original manuscript.
The response to the relationship is unsatisfactory. In WhatsApp, a person can text only for a number already in the friend's list. If a person number is not added in the list and a person texts for the first time then WhatsApp will flag it. The question remains the same that how it will be categorized that a number not added in the list is of a stranger, acquaintances or friends of friends. How the relationship decided? In Figure 4, there are some messaging going back and forth between friends to check the number reputation, what happens if friends are unable to communicate due to networking or any other issue?
A stranger categorisation is straightforward that it is not on the list. If a stranger text for the first time then is the number promoted from stranger to acquaintance upon reply or friend of a friend?
Do the authors categorise the acquaintance as to whom a message is previously sent and the number is promoted from stranger to acquaintance?
Authors need to consider different use cases of those relationships as the whole trust model is dependent on the relationships.
Response 1:
Thank you very much for drawing our attention to the ambiguity of this matter. In WhatsTrust, all of the three relationships: acquaintances, friend of friend and strangers, are not included in the user contact list. Therefore, the algorithm includes a certain check for each case. As shown in section 4.1 in the manuscript, each node maintains a local trust list, which keeps the trust information for all nodes with which he has previously interacted. The information includes the node ID (sender mobile number), positive rating, and negative rating. More details about when a sender is added to a node’s local trust list and how his trust information is updated are provided in the node algorithm (section 4.2.1). The relationships between nodes are determined based on whether the sender node exists in the receiver node’s local trust list and/or contact list or in his friends’ contact lists. Below, we explain the relations in more details.
- Friend: A node n0 considers another node ni (e.g., n2, n3, and n4) a friend if ni’s mobile phone number (ID number) exists in n0’s contact list.
- Acquaintance: A node n0 considers another node (e.g., n7) an acquaintance if n7 exists in n0’s local trust list but is not in the contact list. Acquaintances may also include group members.
- Friend of a friend (FoF): A node n0 considers another node (e.g., n6) a FoF if n6 does not exist in n0’s local trust list or contact list, but receive a response from his friends indicating that he is in their contact lists. As explained in section 4.2.1, when a node n0 receives a message from n6 for the first time, n0 checks with his friends (e.g., n4) if n6 is in their contact list. If so, n0 considers n6 to be a FoF.
- Blocked: A node n0 may block another node (e.g., n1), which prevents node n1 from sending messages to n0.
- Stranger: In all other cases, a node n0 considers other nodes (e.g., n5) to be strangers. This indicates that n5 is not in n0’s contact list or local trust list, nor he is in his friends’ contact lists.
In case a sender does not exist in the node’s contact list or local trust list, the node communicates with his friends to check if the sender is in their contact list (i.e., a FoF) or not (i.e., a stranger). The node waits for a response from his friends for a predetermined period of time. If no response is received because they do not know the node or for any other communication issues, the node considers the sender a stranger.
To address this comment, the following revisions are made to the manuscript:
- The relationship models in section 3.3 are revised to incorporate more details as described above.
- The following sentence is added to the node algorithm (section 4.2.1) line 348 to address the comment about communication issues, “Third, if no friend responds within a predetermined period of time because they do not know or for any other communication issues, is considered a stranger.”

Round 3
Reviewer 1 Report
The proposed solutions to keeping a local trust list would create privacy issues. However, this could be addressed in future work as to how to anonymise the contacts in the local trust list stored somewhere in a local or central database.
The reviewer is satisfied with the author's response and believes that this research work would be a good contribution to the research literature.